# HOMESAFEBENCH: A BENCHMARK FOR EMBODIED VISION-LANGUAGE MODELS IN FREE-EXPLORATION HOME SAFETY INSPECTION

## ABSTRACT

Embodied agents can identify and report safety hazards in the home environments. Accurately evaluating their capabilities in home safety inspection tasks is curcial, but existing benchmarks suffer from two key limitations. First, they oversimplify safety inspection tasks by using textual descriptions of the environment instead of direct visual information, which hinders the accurate evaluation of embodied agents based on Vision-Language Models (VLMs). Second, they use a single, static viewpoint for environmental observation, which restricts the agents' free exploration and cause the omission of certain safety hazards, especially those that are occluded from a fixed viewpoint. To alleviate these issues, we propose HOMESAFEBENCH, a benchmark with 12,900 data points covering five common home safety hazards: fire, electric shock, falling object, trips, and child safety. HOMESAFEBENCH provides dynamic first-person perspective images from simulated home environments, enabling the evaluation of VLM capabilities for home safety inspection. By allowing the embodied agents to freely explore the room, HOMESAFEBENCH provides multiple dynamic perspectives in complex environments for a more thorough inspection. Our comprehensive evaluation of mainstream VLMs on HOMESAFEBENCH reveals that even the best-performing model achieves an F1-score of only 10.23%, demonstrating significant limitations in current VLMs. The models particularly struggle with identifying safety hazards and selecting effective exploration strategies. We hope HOMESAFEBENCH will provide valuable reference and support for future research related to home security inspections. Our dataset and code will be publicly available soon.

## 1 INTRODUCTION

Homes often present safety hazards due to human negligence, potentially posing a serious threat to residents (Stewart, 2001; Josephson et al., 1991; Goldstick et al., 2022). While regular inspections can prevent these issues, manual checks are time-consuming and labor-intensive. Fortunately, the recent development of vision language models (VLMs) (Alayrac et al., 2022; Liu et al., 2023; Wang et al., 2024; Bai et al., 2025a) has enabled VLM-based embodied agents to perform various practical tasks such as visual exploration, navigation, and embodied question-answering (Duan et al., 2022; Chen et al., 2019; Batra et al., 2020; Ye et al., 2021; Zhao et al., 2025). The embodied VLM agents show great promise for diverse applications, especially within home environments (Yin et al., 2024; Liu et al., 2024). Consequently, the automation of safety inspections using embodied VLMs agents is a promising new area of research.

However, the evaluation of embodied VLM agents in home safety inspection tasks still has significant flaws. Specifically, previous evaluation benchmarks exhibit notable limitations (Mullen Jr et al., 2024; Hassan et al., 2024) primarily in two aspects. First, they convert visual data into textual modalities such as object relationship graphs, for processing by text-only large language models (LLMs). This modality transformation discards critical spatial information, as nuanced spatial concepts are simplified into inadequate positional relationship descriptions in text, thus failing to evaluate the general visual understanding capabilities of VLM-based embodied agents. Second, they rely on fixed-view cameras for hazard identification. The fixed and limited field-of-view is susceptible to occlusion, potentially causing the embodied VLM agents to overlook hazards.

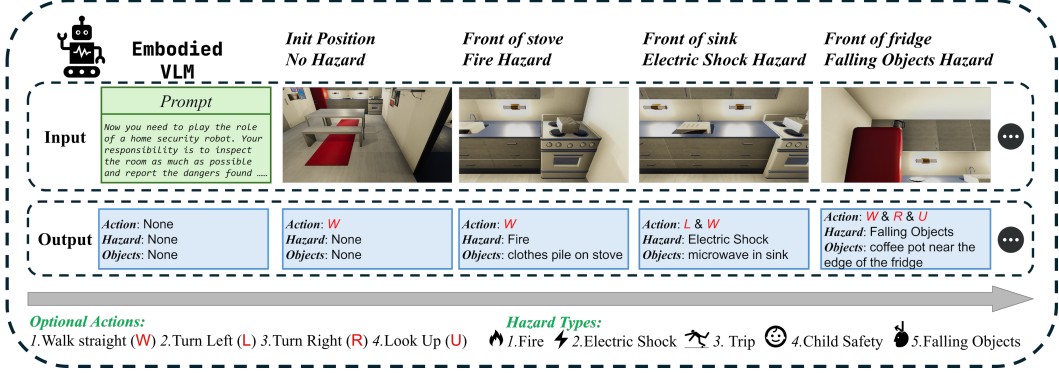

Figure 1: Schematic diagram of the home safety inspection. VLM agents are tasked with identifying the objects that pose a safety hazard given the first-person perspective image from the environment, and select the next action from the action list to iteratively inspect the entire room.

To address the lack of visual presentations and flexible viewpoints in existing home inspection benchmarks, we propose HOMESAFEBENCH, a comprehensive benchmark for evaluating safety inspection performance of embodied VLMs under free exploration with visual feedback. The construction of HOMESAFEBENCH combines human annotation and rule-based generation, obtaining a large scale dataset with significant diversity. Throughout and after the construction process, human reviews are adopted to ensure the correctness and high quality of the benchmark. Collectively, HOMESAFEBENCH contains 12,900 safety inspection tasks with diverse variations based on the simulated environment of VirtualHome (Puig et al., 2018; 2020), covering five prevalent domestic hazard categories: fire, electric shock, falling object, trip, and child safety. Each instance represents a room environment containing multiple hazards, and agents are instructed to autonomously explore the room to report these hazards. During the inspection, the VLM-based agents interacts with the environment to acquire egocentric visual perspectives, identifies hazards within the current field-of-view, and autonomously determines subsequent actions. The process is shown in Figure 1.

We conduct a systematic evaluation on mainstream VLMs using HOMESAFEBENCH. Our results show that existing VLMs have significant capability deficiency in identifying potential home safety hazards under the paradigm of free exploration with visual feedback. Even the top-performing proprietary VLMs like Qwen-VL-Max and GPT-4o achieve an F1 score under 10%. Our finding indicates that current VLMs are not yet reliable for real-world applications in home safety inspection.

To offer a deeper understanding of the deficiencies of current VLMs in home safety inspection tasks, we conduct an in-depth analysis of the effectiveness of free exploration by embodied VLM agents. Our key findings highlight the importance of free exploration in these tasks, while also revealing a significant weakness in the exploration effectiveness of current embodied VLMs, particularly in complex environments and over a larger number of interaction steps.

Our main contributions are as follows:

- We introduce HOMESAFEBENCH, a novel benchmark for embodied VLMs on home safety inspection, which enables visual feedback and agentic free exploration. HOMESAFEBENCH contains 12,900 instances across five categories of common household safety hazards, including fire, electric shock, falling object, trip, and child safety hazards. Its quality is ensured through carefully designed construction process and extensive reviews.

- We conduct a comprehensive evaluation of prevalent VLMs using HOMESAFEBENCH. Our results show the significant limitations of existing models in home safety inspection tasks under a free exploration paradigm with visual feedback, demonstrating that HOMESAFEBENCH is a highly challenging benchmark.

- We conduct an in-depth analysis of VLM agents' free exploration during safety inspections to understand the root of their deficiencies. We demonstrate that while free exploration is crucial for a successful inspection, it remains a significant challenge for VLM agents, especially in complex environments and over a larger number of interaction steps.

## 2 RELATED WORK

### 2.1 VISION-LANGUAGE MODEL ON EMBODIED AI

The superior performance of Language Models (LMs) on various downstream tasks has motivated researchers to explore their application in embodied AI. In early work, researchers attempt to convert the image modality into the text modality to adapt to Large Language Models (LLMs) (Liu et al., 2024; Zhu et al., 2024). However, this approach inevitably leads to loss of image information. With the rise of Vision-Language Models (VLMs) (Bai et al., 2025a; Chen et al., 2024b;a), using VLMs to process the image modality in embodied AI has become mainstream.

VLM are widely used in embodied AI scenarios (Ma et al., 2024; Li et al., 2025b; Shao et al., 2025), where a VLM-based agent gathers information about the surrounding environment through interaction and updates its internal environment models. In vision-language navigation tasks, embodied VLMs are tasked with conduct navigation given natural language commands (Anderson et al., 2018; Zhu et al., 2020; Hong et al., 2021; Zhang et al., 2025). Embodied question answering tasks (Das et al., 2018; Ong & Jang, 2025; Li et al., 2025c; Zhao et al., 2025) require agents to actively explore in an environment to collect visual information and answer natural language questions about the environment. Embodied VLM agent task involves robots performing complex and language-driven physical tasks, ranging from single turn instruction following, to long-horizon planning Gao et al. (2022); Li et al. (2023); Patel et al. (2025); Yang et al. (2025); Sripada et al. (2025).

Our proposed HOMESAFEBENCH introduces a challenging multi-task scenario in the embodied VLM tasks, which is to discover safety hazards in the room as much as possible, and complete the room safety inspection. Such home safety inspection task requires not only the VLM to identify safety hazards, but also to independently decide the next action with the space perception and planning capability. Compared to existing work, HOMESAFEBENCH introduces a novel and challenging task that is of great practical use.

### 2.2 SAFETY OF EMBODIED AI

With the extensive development of embodied AI, its safety has become a critical direction requiring researchers' attention. In recent years, numerous studies on embodied AI safety have emerged, broadly falling into two categories: making embodied AI itself safer (Yin et al., 2024; Liu et al., 2024; Zhang et al., 2024a; Zhu et al., 2024; Zhang et al., 2024b) and utilizing embodied AI to accomplish human safety-related tasks (Li et al., 2025a; Zhou et al., 2024; Mullen Jr et al., 2024; Hassan et al., 2024). For the task of making embodied AI safer, researchers typically focus on whether the operations performed by the agent are safe. For instance, techniques like prompt injection or jailbreaking are used to enable the model to execute dangerous human instructions normally (Yin et al., 2024; Zhang et al., 2024a) or generate dangerous specific actions (Zhu et al., 2024; Zhang et al., 2024b). In scenarios where embodied AI performs human safety-related tasks, it is often required to address hazardous situations in real life, such as analyzing the causes of traffic accidents (Li et al., 2025a), rescuing items in fire, flood, or strong wind environments (Zhou et al., 2024), or checking homes for unsafe or unsanitary conditions (Mullen Jr et al., 2024).

The goal of our work is to utilize embodied AI to conduct home safety inspection, belonging to the second category of accomplishing human safety-related tasks. Compared to previous work, our embodied environment poses a significantly more flexible and challenging scenario. Specifically, the agents are required to autonomously patrol the home to identify safety hazards with the visual feedback, placing great demands on the agent's spatial conception and path planning capabilities. The detailed comparison is shown in Table 2.

## 3 HOMESAFEBENCH

### 3.1 TASK DEFINITION

We propose a home safety hazard inspection task in which an embodied agent actively navigates a simulated 3D home environment to identify and report safety hazards. Following real-world home safety guidelines, we define five categories of common household hazards in our benchmark. Each category represents a specific configuration of item placement that poses a safety risk.

- **Fire Hazards**: Flammable materials are located close to active or potential heat sources. Examples include curtains or stacks of paper placed next to a lit stove, and a pile of dry cloth near a burning candle.
- **Electric Shock Hazards**: Appliances or power devices in contact with water, which may cause electric shock or short circuits. Examples include an appliance in a sink or a toilet.
- **Falling Object Hazards**: Items positioned in a way that they may fall from height and cause injury or damage. Examples include a coffee pot placed at the edge of a refrigerator, or a box positioned at the edge of a shelf.
- **Trip Hazards**: Objects or clutter on the floor that could cause someone to stumble or lose balance during normal movement. Examples include a bar of soap left in a hallway.
- **Child Safety Hazards**: Placement of dangerous or harmful items within easy reach of a child. Examples include a bottle of alcohol on a low table, or sharp kitchen knives placed on TV stand.

Formally, let the initial state denoted as $s_0$ with a ground-truth hazard set $\mathcal{H}$. At each discrete time step $t$, the agent policy $\pi$ is to identify hazards $\hat{\mathcal{H}}_t$ with the current observation, and select an action $a_t \in \mathcal{A}$. The state is then transitioned following the transition function $f$, updating the observation.

$$\hat{\mathcal{H}}_t, a_t \sim \pi(\cdot|s_t), \quad s_{t+1} = f(s_t, a_t). \tag{1}$$

The action space $\mathcal{A}$ consists of basic navigation primitives such as `move-forward`, `turn-left`, `turn-right`, and `look-up`, as detailed in Appendix A. After executing a sequence of actions $\{a_0, a_1, \ldots, a_{T-1}\}$ within a step budget $T$, the final identified hazard set is defined as the union of the hazard sets identified at each step.

$$\hat{\mathcal{H}} = \bigcup_{i=0}^{T-1} \hat{\mathcal{H}}_t. \tag{2}$$

Task performance is evaluated by comparing the reported hazards $\hat{\mathcal{H}}$ against the ground-truth hazards $\mathcal{H}$ using the precision, recall, and F1 score. A hazard is considered correctly reported if both its category and the name of the associated item are correct. Note that we do not require a perfect match for the item name. Instead, we use a rule-based matching system for a more flexible and reliable evaluation, as detailed in Appendix A.

$$\text{Precision}(\hat{\mathcal{H}}, \mathcal{H}) = \frac{||\hat{\mathcal{H}} \cap \mathcal{H}||}{||\hat{\mathcal{H}}||}, \quad \text{Recall}(\hat{\mathcal{H}}, \mathcal{H}) = \frac{||\hat{\mathcal{H}} \cap \mathcal{H}||}{||\mathcal{H}||}, \quad \text{F1}(\hat{\mathcal{H}}, \mathcal{H}) = \frac{2 \times ||\hat{\mathcal{H}} \cap \mathcal{H}||}{||\hat{\mathcal{H}}|| + ||\mathcal{H}||}. \tag{3}$$

## 3.2 CONSTRUCTION AND QUALITY CONTROL PROCEDURE

HOMESAFEBENCH is constructed based on the engine of VirtualHome by combining manual annotation and rule-based generation. The careful reviews are conducted through out the construction process to ensure the correctness and quality of the benchmark. The construction process consists of three stages, as shown in Figure 2.

**Annotation of Potential Hazard Locations** Firstly, the spatial locations within the virtual environment that are likely to contain safety hazards are annotated, such as the top of a refrigerator, inside a sink, or on a stove. The process is performed by two annotators, each responsible for six rooms across three environments (12 rooms in total). To ensure annotation quality, the annotators cross-verify each other's annotation case by case, filter out locations with low risks, and the final annotations reflect consensus between the annotators. Each identified location is assigned exactly one hazard type tag. In total, we obtain 136 annotated hazard locations across all environments.

**Annotation of Object Attributes** Then, the common objects in the virtual environment are assigned with a predefined set of attributes, including flammable, electrical, tripping hazard, falling object, and child safety hazard. An object may be assigned multiple attributes. In total, 367 objects across the three environments are separately annotated by two annotators, and any disagreement between them is referred to a third annotation for a consolidated assignment.

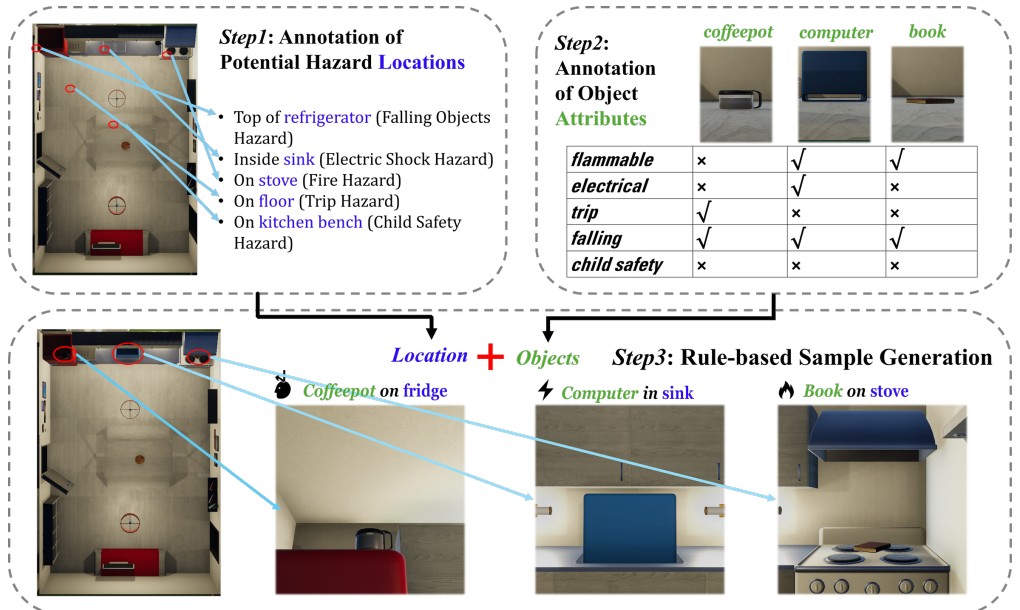

Figure 2: Annotation process of HOMESAFEBENCH.

**Rule-based Sample Generation** Finally, the final samples of HOMESAFEBENCH are generated following the combination rule of locations and objects. Specifically, based on the potential hazard types associated with these locations, we place suitable objects with corresponding attributes at the sampled positions. For instance, a location tagged as fire hazard (e.g., a stove) may have an object of paper placed on it, while a location tagged as falling object hazard (e.g., the top of a refrigerator) may have a glass cup assigned to it.

For quality control of the final samples, we conduct tests in two ways. First, we randomly select 100 samples, examine every hazard of the gold labels in the virtual environment. It is verified that all the hazard points marked are indeed risky and the placement of items is visually discoverable. Second, we randomly select other 100 sample, and manually conduct inspection with no golden label given. The Precision, Recall, and F1 scores achieve 82.29%, 69.50%, and 75.36% for human inspection, validating that the tasks in HOMESAFEBENCH are solvable.

## 3.3 DATASET STATISTICS

Collectively, HOMESAFEBENCH contains 12,900 samples generated from 12 unique scenes, covering five types of hazards. Considering its relatively large size, we select a small subset sized 1,580 as detailed in Appendix A for quick experiments. Hazard statistics are shown in Table 1.

Table 1: Statistics of hazard types in HOMESAFEBENCH for subset (top) and full set (bottom).

|  | Fire | Electric Shock | Trip | Falling object | Child Safety |
|---|---|---|---|---|---|
| # of Hazard Locations | 17 | 3 | 51 | 13 | 19 |
| # (and %) of Samples | 958 (60%) | 417 (26%) | 1486 (94%) | 973 (61%) | 716 (45%) |
| # Hazard Locations | 22 | 10 | 58 | 24 | 22 |
| # (and %) Samples | 9051 (70%) | 3558 (28%) | 11019 (85%) | 8836 (69%) | 8999 (70%) |

Compared to existing safety inspection datasets in Table 2, HOMESAFEBENCH is larger in scale. More importantly, it is built within interactive virtual environments, enabling visual feedback and free exploration of embodied VLMs.

Table 2: Comparison of HOMESAFEBENCH with existing safety related datsets, including Safety-Detect (Mullen Jr et al., 2024), M-CoDAL (Hassan et al., 2024), SafeAgentBench (Huang et al., 2025), and Safe-BeAl (Huang et al., 2025).

| Dataset | Samples | Hazard Categories | Scene | Visual | Free Exploration |
|---|---|---|---|---|---|
| SafetyDetect | 1,000 | 3 | 7 | ✗ | ✗ |
| M-CoDAL | 908 | 16 | ✗ | ✔ | ✗ |
| SafeAgentBench | 750 | 10 | 1 | ✔ | ✗ |
| Safe-BeAl | 2,027 | 8 | 1 | ✗ | ✗ |
| **HOMESAFEBENCH** | 12,900 | 5 | 12 | ✔ | ✔ |

## 4 EXPERIMENTS

### 4.1 SETTINGS

**Models**    Considering the information loss during image-to-text conversion, we chose VLMs over LLMs as the foundation models for embodied home safety inspection agents. We comprehensively test mainstream VLMs. Open-sourced models are locally deployed, including Qwen2.5-VL-7B (Bai et al., 2025b), InternVL2.5-4B, InternVL2.5-8B (Chen et al., 2024b), Llama3.2-11B-V (Dubey et al., 2024), and Gemma3-12B (Team et al., 2025). proprietary models are called through API, including Qwen-VL-Max (Bai et al., 2023) and GPT-4o (Hurst et al., 2024).

**Inference**    The open-sourced models are locally deployed with `transformers` (Wolf et al., 2020). We set temperature as $0.6$, top-p as $0.9$ during sampling for all models. We don't use greedy decoding to avoid the endless repetition of the generated actions.

**Agent Design**    The interaction flow between the VLM agents and the virtual environments is illustrated in Figure 1. The agents perform a 10-turn dialogue-based room inspection, where environment transmits a first-person perspective image to the VLM, and the VLM identifies and reports safety hazards based on the image, then autonomously decides the next action. Agent prompts, and other implementation details are listed in Appendix B.

**Metrics**    We report the micro average of precision, recall, and F1 scores as the final metrics following Equation 3. Specifically, we calculate the metrics for each task instance, and average across the dataset sourced from HOMESAFEBENCH.

### 4.2 MAIN RESULT

Table 3: Main results of embodied VLMs on HOMESAFEBENCH. Best scores among all models are shown in bold. Prec and Rec refer to Precision and Recall respectively.

| Models | Subset | | | Others | | | All | | |
|---|---|---|---|---|---|---|---|---|---|
| | **Prec** | **Rec** | **F1** | **Prec** | **Rec** | **F1** | **Prec** | **Rec** | **F1** |
| Qwen2.5-VL-7B | 5.16 | 2.42 | 2.91 | 1.61 | 0.87 | 1.02 | 1.97 | 1.03 | 1.21 |
| InternVL2.5-4B | 7.98 | 2.73 | 3.66 | 4.10 | 1.25 | 1.69 | 4.57 | 1.43 | 1.93 |
| InternVL2.5-8B | 9.38 | 3.05 | 4.06 | 7.42 | 1.91 | 2.87 | 7.67 | 2.05 | 3.01 |
| Llama3.2-11B-V | **9.77** | 17.84 | **11.84** | 7.87 | 11.11 | 8.78 | 8.11 | 11.93 | 9.16 |
| Gemma3-12B | 9.00 | **18.14** | 11.46 | 8.51 | **12.93** | **10.06** | 8.57 | **13.57** | **10.23** |
| Qwen-VL-Max | 7.26 | 3.15 | 4.03 | 5.13 | 1.64 | 2.23 | 5.15 | 1.76 | 2.44 |
| GPT-4o | 7.67 | 4.44 | 5.42 | **10.30** | 6.15 | 6.89 | **9.91** | 5.89 | 6.67 |

In the main experiment, we evaluate VLMs using the complete HOMESAFEBENCH dataset. The results are shown in Table 3. In safety hazard identification, all models scores below 20% on Precision, Recall, and F1. Comparing to human inspectors that obtaining 82.29%, 69.50%, and 75.36%

Precision, Recall, and F1 on a subset, it can be concluded that current VLMs have a very poor performance on home safety inspection tasks. Even the commercial models Qwen-VL-Max and GPT-4o, which perform well on many tasks, achieved low scores comparable to smaller models. Gemma3-12B achieved the best performance among all, with a recalling 13.57% of all hazards. Qwen2.5-VL-8B achieved the lowest score, whose action selection is single-minded, usually selecting and repeatedly executing only one action. Furthermore, Qwen2.5-VL-8B was more likely to choose the passive "None" answer rather than proactively identifying safety hazards. To further understand the poor performance of current VLMs, we conduct a analysis on their free exploration behaviors in Section 5, and introduce case studies in Appendix D.

As detailed in Appendix A, the subset contains more hazard points in obvious places which are easier to notice. Comparison between results of different test sets show that the subset generally obtains higher scores, which meets our expectations. However, the final results are still very low compared to human scores (75.36% F1 score), again validating the tasks of home safety inspection from HOMESAFEBENCH pose strong challenges to VLMs.

## 5 ANALYSIS

In the introduction of HOMESAFEBENCH, we utilize an interactive simulated environment to enable free-exploration for home safety inspection, bridging the gap between evaluation and real-world performance. In the design of VLM embodied agents, we allow the VLM to control the agent's free exploration by generating navigation primitives such as `move-forward`, `turn-left`, `turn-right`, and `look-up`, as detailed in Appendix A. The paradigm of free exploration plays a crucial role in our proposed HOMESAFEBENCH. On one hand, it introduces more flexibility to home safety inspection tasks thus raising its upper-bound capability. On the other hand, it places greater demands on embodied agents and poses greater challenges for VLMs, which potentially leads to the low performance of current models, as shown in Table 3.

To offer a deeper understanding of the role free exploration plays in home safety inspection tasks, we conduct in-depth analysis to answer the three research questions (RQs) about free exploration.

- **RQ1**: Is free exploration useful for home safety inspection task?
- **RQ2**: How is the free exploration performance of current embodied VLMs?
- **RQ3**: How does the multi-turn interaction affect exploration effectiveness?

### 5.1 IMPORTANCE OF FREE EXPLORATION

Table 4: The performance with (w/) and without (w/o) free exploration, and the corresponding difference between the two paradigms ($\Delta$).

| Models | Precision | | | Recall | | | F1 | | |
|---|---|---|---|---|---|---|---|---|---|
| | w/ | w/o | $\Delta$ | w/ | w/o | $\Delta$ | w/ | w/o | $\Delta$ |
| Qwen2.5-VL-7B | 1.97 | 11.09 | -9.12 | 1.03 | 0.41 | +0.62 | 1.21 | 0.79 | +0.42 |
| InternVL2.5-4B | 4.57 | 3.58 | +0.99 | 1.43 | 0.55 | +0.88 | 1.93 | 0.94 | +0.99 |
| InternVL2.5-8B | 7.67 | 4.03 | +3.64 | 2.05 | 0.63 | +1.42 | 3.01 | 1.08 | +1.93 |
| Llama3.2-11B-V | 8.11 | 5.64 | +2.47 | 11.93 | 1.51 | +10.42 | 9.16 | 2.32 | +6.84 |
| Gemma3-12B | 8.57 | 19.17 | -10.60 | 13.57 | 3.41 | +10.16 | 10.23 | 5.69 | +4.54 |
| Qwen-VL-Max | 5.15 | 16.74 | -11.59 | 1.76 | 1.13 | +0.63 | 2.44 | 2.12 | +0.32 |
| GPT-4o | 9.91 | 14.00 | -4.09 | 5.89 | 2.33 | +3.56 | 6.67 | 3.96 | +2.71 |

Intuitively, a fixed single viewpoint of agents can cause problems such as blurring of distant objects and obstruction of safety hazards by other objects. To validate the impact of agent free exploration on inspection capabilities, we design an inspection experiment without it. The experiment is conducted with 10% randomly sampled data of HOMESAFEBENCH, ensuring that each type of room and environment is included. We fix the agent in a corner of the room and rotated it to ensure that the agent's first-person perspective could see the entire room, then calculate the model's Precision, Recall, and F1 scores.

The results are shown in Table 4. When deprived of the ability to explore freely, all models suffer from a consistent and significant decrease in F1 scores. Although Qwen2.5-VL, Qwen-VL-Max, and Gemma3-12B models achieve improvements in precision compared to when deprived, they suffer from a more significant recall drop due to insufficient environmental information and occlusion, finally scoring lower F1. These results indicate that enabling the VLM to control the agent's free exploration is a key factor in ensuring the model's effectiveness in safety inspection tasks.

## 5.2 PERFORMANCE OF FREE EXPLORATION

Given that free exploration plays a significant positive role in home safety inspection by offering flexible viewpoints, we are interested in the performance of current VLM-based agents in effectively conduct exploration. In doing so, we introduce the Navigation (Nav) metric to reflect the agent's ability to sufficiently navigate the entire environment. Specifically, we use the built-in function `get_visible_objects` in VirtualHome to record the objects visible to the agent, and then calculate the ratio of the number of hazards into the occurrence of the agent view, to the total number of hazards.

$$\text{Navigation} = \frac{||\mathcal{O}_{vis} \cap \mathcal{H}||}{||\mathcal{H}||}, \tag{4}$$

where $\mathcal{O}_{vis}$ refers to the set of all objects visible to the agent, and $\mathcal{H}$ is the ground truth hazard set. The experimental results are shown in Table 5.

Table 5: The navigation (Nav) and F1 scores across different types of rooms. The navigation score of an agent is defined as the ratio of the number of observed hazards to that of all hazards. The highest scores among models are shown in bold.

| Models | Kitchen | | Bathroom | | Bedroom | | Livingroom | | All | |
|---|---|---|---|---|---|---|---|---|---|---|
| | Nav | F1 | Nav | F1 | Nav | F1 | Nav | F1 | Nav | F1 |
| Qwen2.5-VL-7B | 36.77 | 0.70 | 54.12 | 1.24 | 40.98 | 3.73 | 23.73 | 0.79 | 33.64 | 1.21 |
| InternVL2.5-4B | 38.36 | 2.58 | 53.27 | 2.69 | 44.55 | 2.02 | 37.10 | 0.63 | 39.45 | 1.93 |
| InternVL2.5-8B | 37.44 | 3.10 | 53.31 | 5.03 | 42.24 | 2.43 | 40.88 | 2.53 | 40.32 | 2.86 |
| Llama3.2-11B-V | 37.06 | 9.94 | 53.69 | **12.07** | 41.41 | 6.91 | 38.46 | 8.86 | 39.24 | 9.16 |
| Gemma3-12B | 43.72 | 9.21 | **60.78** | 11.18 | 47.33 | **10.78** | 25.22 | **11.68** | 39.54 | **10.23** |
| Qwen-VL-Max | 36.77 | 3.29 | 54.12 | 5.11 | 43.98 | 2.51 | 44.03 | 0.93 | 41.24 | 2.44 |
| GPT-4o | **44.22** | **10.45** | 52.64 | 10.01 | **51.36** | 3.34 | **51.89** | 2.87 | **49.94** | 6.67 |

First, all the models perform badly on navigation, with less than 50% of the risk points included in the observation throughout the inspection. The poor navigation scores partially explain the deficiencies of current embodied VLMs in home inspection task. Since embodied VLMs have significant difficulties navigating to extensively observe the items, it becomes less likely for them to identify the hazards and get a high recall rate.

Second, according to the experimental results, the model performs different on navigation and F1 scores in different rooms. All models perform poorly in the living room, likely due to the large number of items such as sofas and TV tables, which complicates the environment and presents a greater challenge for the models. In the bathroom, most models achieve relatively strong scores, likely due to the smaller size and fewer items of the room which makes it easier for the models to inspect the entire room. The difference between rooms indicate that current VLM agents still struggle with conducting effective navigation in complex environments.

## 5.3 FREE EXPLORATION UNDER MULTI-TURN INTERACTION

The free exploration in HOMESAFEBENCH involves multi-turn interaction with the virtual environment. A natural question under this paradigm is, how will the effectiveness of free exploration be affected as the number of turn grows. Therefore, we conduct an investigation by setting the maximum number of turns to 30, and calculating the model score every five turns. This maximum number of turns is set to 30, as empirical evidences suggest that the models tend to output meaningless content, and the performance will not change significantly when the number of turns exceeds 30. The

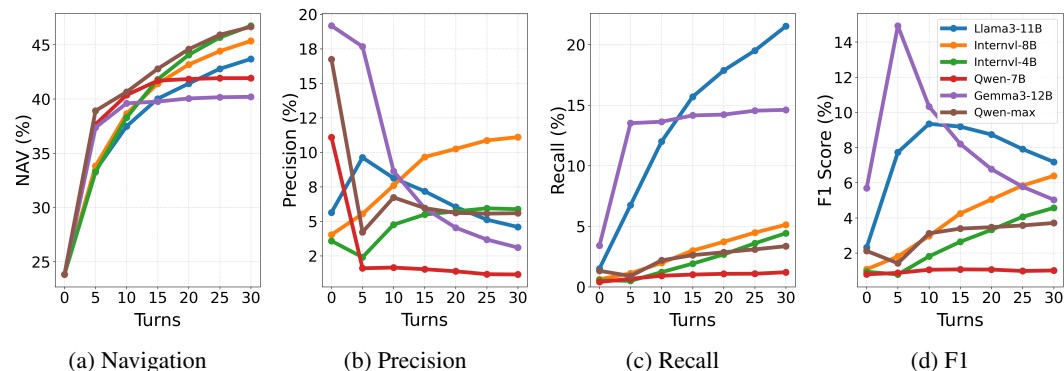

|              |                |              |              |
|--------------|----------------|--------------|--------------|
| (a) Navigation | (b) Precision | (c) Recall | (d) F1 |

Figure 3: VLM performance changes as a function of the number of action turns. Sub-figure (a), (b), (c), and (d) shows the change of Navigation score, Precision score, Recall score, and F1 score of each VLM with action turns.

model's Navigation, Precision, Recall, and F1 scores as a function of the number of turns as line graphs are shown in Figure 3.

As is demonstrated in Figure 3, as the number of interaction turn grows, the F1 score does not necessarily increases monotonically with it. Instead, the performance general reaches its high value or even its peak at the first few steps, and then saturates or even decrease along with more steps. Analysis on other metrics other than F1 scores offers further insights about the long-horizon free exploratino process. Along with the turn number increases, the Navigation score and Recall score gain per five turns gradually slows down, indicating that the free exploration and hazard identification are less and less effective for later turns in the whole process. Meanwhile, the Precision score significantly decreases when turn number grows, also demonstrating the decrease of inspection quality after a certain number of turns.

Generally, the analysis highlights another weakness of current embodied VLMs in the free exploration of home safety inspection tasks, namely the effectiveness drop under a long horizon of multi-turn interaction. These VLM-based agents lack clear and solid planning to conduct a well-organized inspection, but conduct exploration and identification in a arbitrary way, thus obtaining little gain in a long-sequence task.

## 6  CONCLUSION

To address the limitations of fixed viewpoints and the loss of critical visual information during home safety inspection evaluation, we propose HOMESAFEBENCH, a benchmark for embodied VLM evaluation on home safety inspection tasks, featured by first-person visual perception and interactive free exploration. HOMESAFEBENCH comprises a large-scale and multi-category dataset constructed based on the VirtualHome simulation environment, containing 12,900 samples with significant variations across five common hazard categories, including fire, electric shock, falling objects, falls, and child safety hazards. The correctness and quality of HOMESAFEBENCH is ensured by human reviews throughout the annotation process.

Using HomeSafeBench, we systematically evaluate the mainstream VLMs, highlighting significant shortcomings of existing models in safety hazard identification and exploration strategies. In particular, we conduct a in-depth analysis on the free exploration pattern of embodied VLMs on home safety inspection tasks. The analysis reveals not only the importance of free exploration, but also the significant weaknesses of VLMs in effective navigation especially in complex environment under a multi-turn paradigm, partially explaining the deficiency of them to conduct safety inspection.

Focusing on the home safety inspection task, our work provide a solid foundation through the comprehensive benchmarking of embodied VLMs. On a broader impact, our key findings indicate the weaknesses of current VLMs on purposeful navigation and hazard identification, and can inspire future work for a general VLM capability improvement.

## 7 ETHICS STATEMENT

Our work falls under the category of embodied agents controlled by VLMs, a field that carries certain inherent risks. However, as a benchmark built upon existing theory and application, our study introduces little additional risks beyond current ones. Moreover, this work proposes HOMESAFEBENCH specifically for security checks, which aims to enhance the safety of embodied VLM systems and thus has a substantial positive impact.

## 8 REPRODICIBILITY STATEMENT

A comprehensive description of our dataset construction process is provided in Section 3 and Appendix A . The experimental setup and implementation details are elaborated in Section 4 and Appendix B for reproduction. Furthermore, our dataset and source code will be publicly available to ensure reproducibility.

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

## A    BENCHMARK DETAILS

### A.1    VIRTUAL ENVIRONMENT

We utilize VirtualHome as the virtual home environment, and the agent within this environment is simulated using the "Character" module from VirtualHome. Specifically, an agent can be added using the `add_character` function. We adopt the `FIRST_PERSON` camera of the "Character" as the primary egocentric viewpoint for all actions except for the "Look Up" action.

Regarding the actions described in Section 3.1, we have modified the original actions provided by VirtualHome as follows:

**Walk Straight**    In VirtualHome, the "walkforward" action moves the agent forward by only one step per execution. However, a single step results in minimal change in the agent's field of view, and traversing an entire room would require an excessive number of meaningless steps. To address this, we define "Walk Straight" as moving forward three steps in sequence, i.e., executing three consecutive "walkforward" actions.

**Turn Left & Turn Right**    In VirtualHome, each "turnleft" or "turnright" action rotates the agent by 30 degrees. Similar to the walking action, this would lead to inefficient and repetitive rotations. Therefore, we define "Turn Left" and "Turn Right" as rotating the agent by 90 degrees, achieved by executing three consecutive "turnleft" or "turnright" actions.

**Look Up**    VirtualHome does not natively provide a "look up" action. To implement this, we attach an upward-facing camera, named "up_camera", to the agent during its initialization. The relative position of this camera is set to $Position = [0, 1.5, 0]$ and its rotation to $Rotation = [-15, 0, 0]$. When the agent performs a "Look Up" action, the image captured by the "up_camera" is used.

### A.2    SMALL SUBSET SELECTION

During the location selection phase, annotators identify 57 conspicuous locations, such as tabletops and floors. Subsequently, from all samples, we form a subset comprising entries where over 50% of hazard locations are located in such conspicuous locations. This subset contains 1,586 data instances and is characterized as relatively simpler compared to the complete dataset.

### A.3    EVALUATION DETAILS

To address the issue that objects in the environment may be referred to by multiple names, which can lead to evaluation inaccuracies if a model uses a synonym instead of the standard name defined in the environment, we create a mapping that links alternative names of objects to their canonical names used in the virtual environment.

This mapping is constructed using GPT-4o, which generated preliminary synonym associations based on the standard object names. The initial mappings were then reviewed and refined by two human annotators to ensure relevance and accuracy, resulting in a finalized mapping table. An example of this mapping is provided in the Table 6.

Table 6: Examples of Mapping

| Standard Name | Mapping Names |
|---------------|---------------|
| Wallshelf | Bookshelf, Rack, Shelf |
| Candle | Wax, Light, flame |
| Computer | PC, Laptop, Desktop |
| Toaster | Bread heater, Bread maker, Tost machine |
| Clothespants | Pants, Trousers, Jeans |
| Fridge | Refrigerator |

During evaluation, object names predicted by the model are first mapped to their standard canonical names before scoring is performed.

## B   EXPERIMENT DETAILS

### B.1   INFERENCE DETAILS

During model inference, we utilize the PyTorch framework and the Transformers library, with the versions of the Python libraries aligned with the requirements of each respective model. The inference process is conducted using six NVIDIA TITAN RTX GPUs in parallel. Executing inference on the entire dataset requires approximately 24 to 48 hours.

### B.2   PROMPT

Our experiments used multiple prompts. To ensure reproducibility, this section demonstrates the prompts used.

---

**Prompt for HomeSafeBench**

You are a home security robot responsible for inspecting the room thoroughly and reporting any dangers found. The input image represents your current first-person perspective. Please complete the following two tasks **in sequence** and follow the specified output formats exactly.

Task 1: Identify Safety Hazards
- Detect if there are any safety hazards in your current view.
- Hazard categories to identify: [fire, electric shock, falling object, trip, child safety].
- For each hazard detected, output the hazard category and the name of the object causing it (only the object name).
- If multiple hazards exist, output all of them. Use the format:
  `Task1: {{hazard1, object1; hazard2, object2; ...}}`
  where each `{{hazard, object}}` pair is enclosed in braces and multiple pairs are separated by semicolons.
- If no hazards are detected, reply with:
  `Task1: {{None}}`

Task 2: Choose Your Next Operation
- Select one operation from: [walk straight, turn left, turn right, look up].
- Choose an operation that helps you inspect the room as much as possible without bumping into objects.
- Output format:
  `Task2: {{your next operation}}`

Examples:
Example 1:
Task1: {{fire, stove}}, Task2: {{turn right}}.
Example 2:
Task1: {{slips and falls, box}}, Task2: {{walk straight}}.
Example 3:
Task1: {{falling objects, coffeepot; electric shock, microwave}}, Task2: {{turn right}}.

**Strictly follow the above formats without adding any other content or explanation.**

Your previous action was: {history}. Please complete the two tasks accordingly.

---

Figure 4: Prompt using in HomeSafeBench

In the main experiment, we used HomeSafeBench to evaluate a mainstream VLM. To ensure the VLM fully understood the task, we split the inspection task into Task 1 and Task 2, describing each task's requirements in detail. For ease of evaluation, we specified the output format and required the VLM to strictly adhere to it. We also provided three examples to facilitate contextual learning for the VLM. As shown in Figure 4.

In the Without Action analysis experiment, we deleted Task 2 and made appropriate modifications to the prompt to ensure that the VLM could complete the task, as shown in Figure 5.

---

## Prompt for Without Action

You are a home security robot responsible for inspecting the room thoroughly and reporting any dangers found. The input image represents your current first-person perspective. Please complete the following task and follow the specified output format exactly.

Task: Identify Safety Hazards

- Detect if there are any safety hazards in your current view.

- Hazard categories to identify: [fire, electric shock, falling object, trip, child safety].

- For each hazard detected, output the hazard category and the name of the object causing it (only the object name).

- If multiple hazards exist, output all of them. Use the format:

  `Task: {{hazard1, object1; hazard2, object2; ...}}`

  where each {{hazard, object}} pair is enclosed in braces and multiple pairs are separated by semicolons.

Examples:
Example 1:
Task: {{fire, stove}}
Example 2:
Task: {{slips and falls, box}}
Example 3:
Task: {{falling objects, coffeepot; electric shock, microwave}}

**Strictly follow the above formats without adding any other content or explanation.**

---

Figure 5: Prompt using in without action

### B.3    USAGE OF DATA

In Section 4, we evaluated all models except GPT-4o using the full dataset. In Section 5, a randomly selected 10% subset of the data was used for the experiments in Sections 5.1 and 5.3. The specific data selection and processing procedures for individual experiments are detailed below.

**GPT-4o**    For both the evaluation and analysis experiments involving GPT-4o, we used a randomly selected sample of 100 instances, ensuring the sample included data from all room types.

**Data for the Without-exploration Experiment**    In Section 5.1, we utilized the selected subset of data and adjusted the agent's viewpoint to ensure a complete view of the entire room. The agent's starting position for each data instance was consistent with the initial position used in the Section 4 experiments. The rotation angles applied in each room are specified in Table 7.

Additionally, the camera configuration was adjusted. A camera with a relative position of $Position = [0, 1.5, 0]$ and rotation of $Rotation = [0, 0, 0]$ was attached to the agent, serving as the primary image capture camera.

Table 7: Initialize of Without-exploration

|      | Kitchen        | Bedroom        | Livingroom     | Bathroom       |
|------|----------------|----------------|----------------|----------------|
| Env0 | Turn right 30° | Turn right 30° | Turn left 30°  | Turn right 30° |
| Env1 | Turn right 30° | Turn right 30° | Turn left 60°  | Turn right 30° |
| Env3 | Turn right 30° | Turn right 30° | Turn left 60°  | Turn left 30°  |

## C  DATASET DETAILS

In this section, we employ examples of hazard locations and types to enhance the clarity of the dataset's details.

### C.1  EXAMPLE OF HAZARD LOCATION

HomeSafeBench dataset comprises 3 distinct environments, each containing 4 different room types, resulting in a total of 12 scenarios. During the data construction phase, the process requires the initial annotation of hazard locations. To illustrate the specifics of our annotation methodology, for all 12 scenarios, a number of Hazard locations are selected, ensuring that at least one instance of each type of hazard is included. An example of a hazard location is presented in Figure 6.

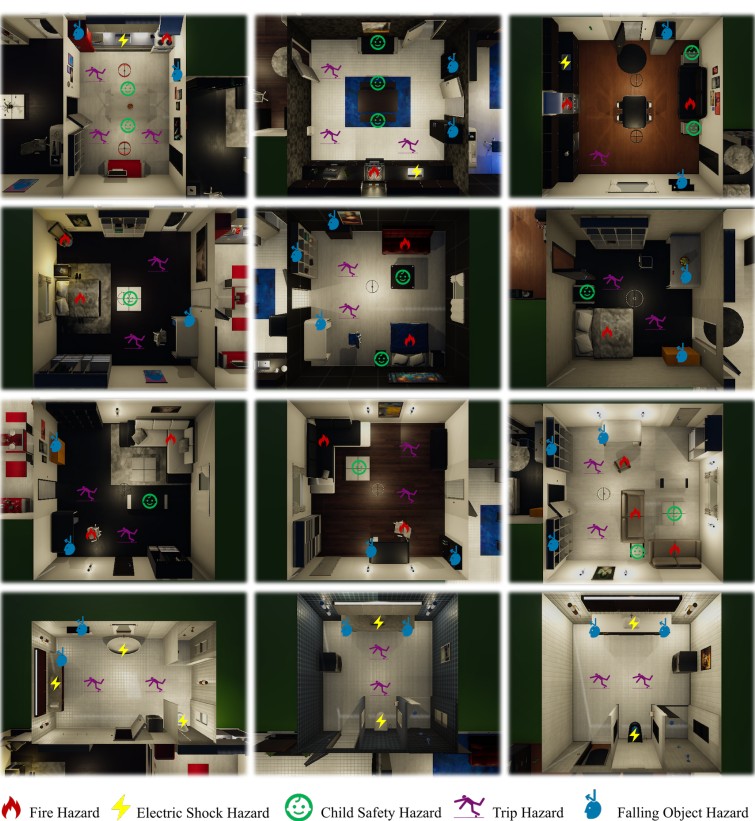

Figure 6: Example of hazard location

### C.2  EXAMPLE OF HAZARD TYPE

HomeSafeBench contains five distinct hazard types, and we provide a representative example for each category, as illustrated in Figure 7.

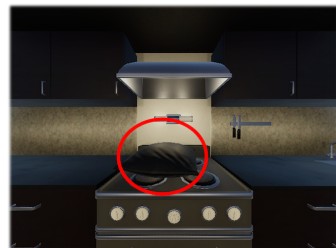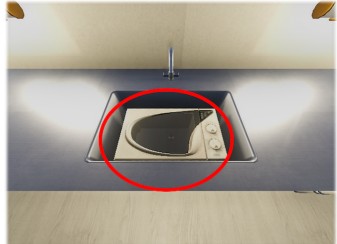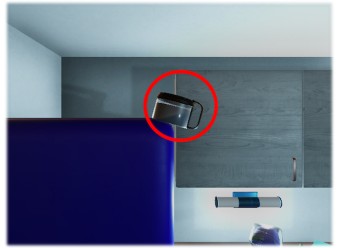

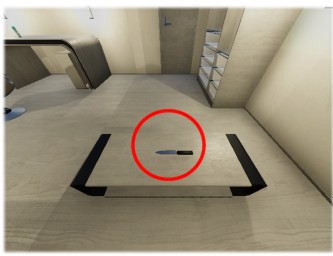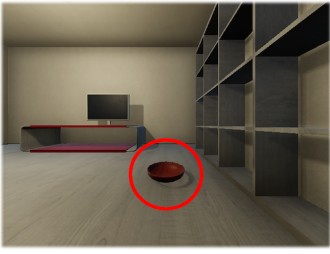

Figure 7: Example of Hazard Type. The left image in the first row illustrates a fire hazard, the center depicts an electric shock hazard, while the right image demonstrates a falling object hazard. The second row presents a child safety hazard on the left and a trip hazard on the right, respectively.

## D  CASE STUDY

To analyze the inaccurate recognition of VLMs, we selected some representative results and analyzed the errors of the model in Navigation and Objects Identification respectively.

**Navigation**  Table 3 and Table 5 shows that all models achieve relatively low recall and Nav scores, suggesting that their suboptimal performance may stem from poor navigation when patrolling rooms. The figure below illustrates several example trajectories of the models operating within the rooms. Figure 8 shows two trajectories produced by GEMMA. In both examples, although the navigation paths differ, the model consistently misses the slip hazard placed on the floor. Notably, the hazard is never observed throughout the entire process.

**Objects Identification**  In addition to navigation errors, models may also produce object recognition mistakes or fail to identify hazards. For example, as shown in Figure 9, even when the model *qwen-max* navigates correctly and observes the hazard *(a computer in the sink, belonging to the **electric shock hazard** category)* within its field of view, it still fails to report it.

## E  USE OF LLMS

The use of LLMs was strictly limited to grammatical corrections and improving readability in the final editing stage, and did not influence the scholarly substance of the work. We guarantee that paper's ideas, writing and experimental settings are solely the work of human authors, did not use LLMs as an assistant.

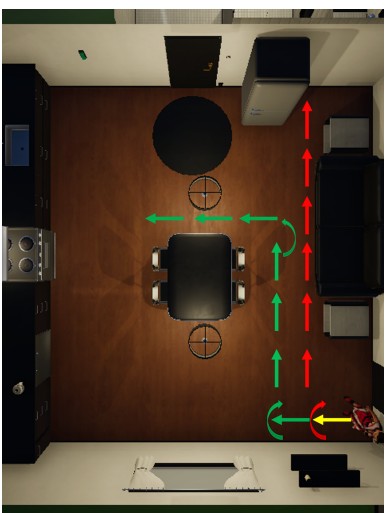

(a) Brief paths (Different colors represent different paths, with yellow indicating the same action).

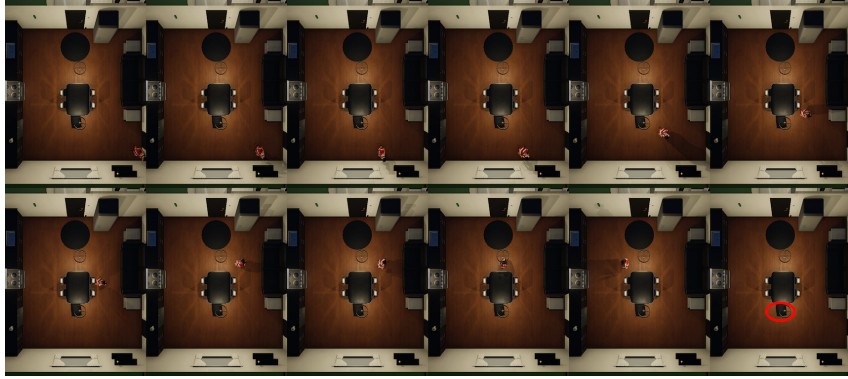

(b) Detailed path (green one in 8a).

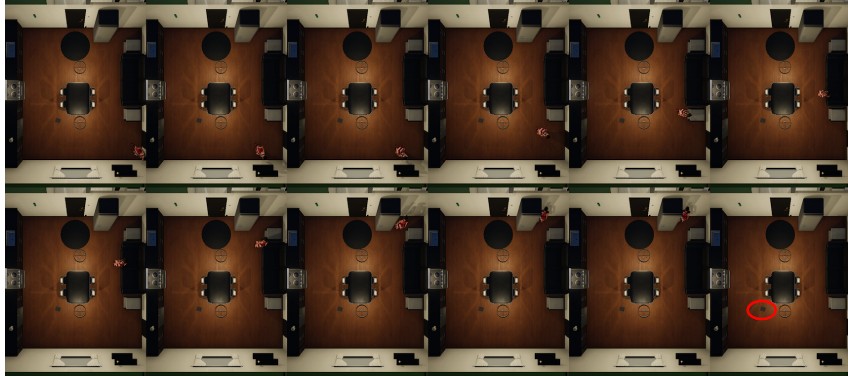

(c) Detailed path (red one in 8a).

Figure 8: Example paths of Gemma3 (Hazards are highlighted with red circles).

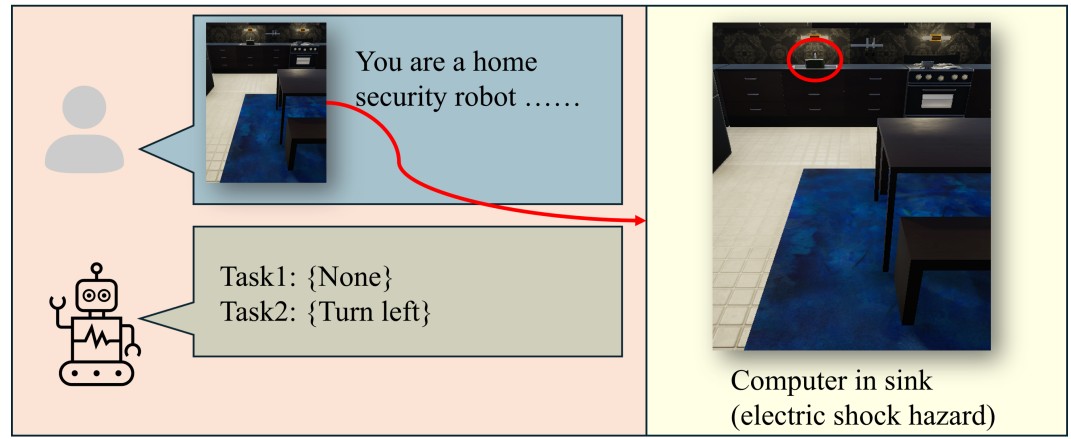

Figure 9: Example of model misidentifications

