# OpenReview forum: "HomeSafeBench: A Benchmark for Embodied Vision-Language Models in Free-Exploration Home Safety Inspection"
_ICLR.cc/2026/Conference — ICLR 2026 Conference Withdrawn Submission_

### Official Review · Reviewer_zaaV · 2025-10-26

**Soundness:** 2
**Presentation:** 3
**Contribution:** 2
**Rating:** 4
**Confidence:** 4

**Summary:**

This work proposes three main contributions:
1. HOMESAFEBENCH, a benchmark for embodied VLMs on home safety inspection, containing 12,900 instances across five categories of common household safety hazards covering fire, electric shock, falling objects, trip hazards, and child safety hazards.
2. The benchmark is used to evaluate open-source VLM models, including Qwen2.5-VL-7B, InternVL2.5-4B, InternVL2.5-8B, Gemma3-12B, Qwen-VL-Max, and GPT-4o on home safety inspection tasks.
3. The paper further conducts an in-depth analysis of VLM agents’ free exploration during safety inspections to understand the root of their deficiencies.

**Strengths:**

1. The main focus of this research is twofold: creating a dataset based on VirtualHome and defining a home safety inspection task. The research ideas and the effort put into home safety inspection are commendable and interesting. The tasks covering five categories of common household hazards are clearly defined and well-motivated.

**Weaknesses:**

1. The modified VirtualHome configuration should be explained more clearly. In the original VirtualHome, movement instructions are generated based on the activity program structure, such as [Walk] <tv> (433), where [action] <object> (object ID) specifies the action and target object. When providing instructions to the agent in your setup, it should be clarified whether these instructions are mainly derived from the prompt or follow a different mechanism.

2. The prompt shows examples of hazard detection and action selection, but it remains unclear how hazards are detected in Task 1 (e.g., rule-based cues, or LLM reasoning) and what principle guides the navigation action selection in Task 2 (e.g., if a safety hazard is detected, is the agent always expected to navigate toward it, avoid it, or continue exploring the environment instead?).

**Questions:**

1. The authors provide a useful comparison of HOMESAFEBENCH with existing safety-related datasets (SafetyDetect, M-CoDAL, SafeAgentBench, and Safe-BeAl) in Table 2. This comparison is valuable, but it would be helpful if the authors clearly indicate whether these existing datasets are synthetic or real-world, as this distinction affects the evaluation context and the significance of the benchmark.

2. Following up on Question 1, the authors should consider whether comparing synthetic and real-world datasets is appropriate. While such datasets can be compared, the context matters: synthetic datasets (e.g., virtual environments) are fully simulated, with precise annotations, controlled scenarios, and flexible configurations, but they may lack the complexity, noise, and variability of real-world environments. In contrast, real-world datasets capture actual sensor and camera data with realistic lighting, clutter, occlusions, and unpredictability, though annotations may be noisier. Clarifying this distinction would help readers better interpret the comparison in Table 2.

3. Please provide a clear comparison between your modified VirtualHome environment and the original VirtualHome. It would be helpful to describe what changes were made, such as differences in action execution, camera configuration, object interaction, or instruction generation, and how these changes affect task evaluation.

4. The authors note that model performance varies across different rooms. For example, environments with more complex layouts, such as living rooms with multiple tables, pose greater challenges for navigation and inspection. In contrast, smaller and simpler rooms, like bathrooms, tend to yield stronger model performance due to fewer objects and easier full-room inspection. This highlights that current VLM agents still struggle with effective navigation in complex environments. However, in real-world scenarios, rooms contain numerous objects, including small or partially occluded items, which need to be detected and inspected. In this work, the authors did not utilize or discuss object detection models such as YOLO, DINO, or SAM, which often serve as fundamental components within VLM pipelines. For example, in autonomous driving systems, most existing approaches first apply object detection and then use VLMs to reason about near-miss accidents, providing captions or descriptions of accident-causing agents, suspect objects, accident types, risk levels, and so on. Therefore, the inspection task presented in this work could be improved by researching and integrating such detection-based approaches.

5. VirtualHome simulations might require high-quality graphics rendering, which can be memory-intensive and demand substantial GPU resources. Additionally, integrating VLMs introduces further computational overhead, particularly when processing high-resolution images or multiple frames sequentially. As a result, execution time may increase significantly, especially for complex scenes with many objects. For practical deployment, it would be helpful if the authors reported the hardware configuration (e.g., GPU type, CPU, RAM), memory usage, and average execution time per frame. This information would allow readers to better assess the feasibility and scalability of the approach in real-world or resource-constrained scenarios.

6. The authors mention that VLMs are employed locally, which is fine. When GPT-4o is run via an API, there is a concern that personal faces or other private information could be exposed. In the current work, using virtual agents, this is not an issue. However, if the approach were applied to real-world data, such as private home data, it would be important to consider strategies for protecting privacy. Do the authors have any ideas for privacy protection?

7. The authors could provide more details about the camera settings, such as top view, ego view, or fixed viewpoints. Extending or varying the camera setup could improve the generality and evaluation of the approach. For reference, similar work has been done in 1) activity scenario simulation by discovering knowledge through daily living activity datasets, https://www.tandfonline.com/doi/pdf/10.1080/18824889.2024.2318848 and 2) the Knowledge Graph Reasoning Challenge for Social Issues (KGRC4SI) https://github.com/KGRC4SI/DataSet, which is publicly available.

---

### Official Review · Reviewer_1HzY · 2025-10-31

**Soundness:** 3
**Presentation:** 3
**Contribution:** 2
**Rating:** 4
**Confidence:** 3

**Summary:**

The paper introduces HomeSafeBench, a new benchmark for evaluating VLMs in a highly practical task, which is home safety inspection.

Contributions:

- Unlike older benchmarks that use fixed cameras or just text descriptions, this benchmark requires agents to autonomously move and explore a 3D environment.
- The dataset is quite large and contains 12,900 instances covering common household safety hazards, ensuring a large variety.
- The paper shows that state of the art VLMs perform quite poorly, showing that the highest F1 score is only 10.23%.
- The paper’s analysis shows that VLM agents perform poorly at multi-step exploration in complex environments.

**Strengths:**

- Significant Advancement in Task Design: The benchmark successfully addresses two major limitations in existing work: eliminating the use of textual object relationship graphs (which discard crucial spatial information) and moving past static viewpoints. By demanding first-person perspectives and free exploration, it creates a far more realistic and challenging scenario for embodied agents.
- Empirical Significance: The evaluation provides a clear result: the best-performing current VLM achieves an F1-score of only 10.23%. This failure is a significant finding that demonstrates a fundamental gap in current models and validates the benchmark's difficulty and necessity.
- Robust Dataset: The dataset is substantial, consisting of 12,900 instances. The construction process utilized a combination of rule-based generation and human cross-verification for hazard locations and object attributes, suggesting a commitment to data quality and correctness.
- Practical Application: The focus on five common household safety hazards gives the work a high degree of practical utility.
- Insightful Analysis: The paper provides analysis that goes beyond simple performance scores, particularly demonstrating the importance of free exploration while simultaneously revealing the significant weakness in the effectiveness of exploration in current VLMs.

**Weaknesses:**

- Limited Generalizability Due to Simulator Choice: The benchmark is constructed using the VirtualHome simulator. The findings may be specific to VirtualHome's limited graphics and object interactions and may not generalize to real-world robotics.
- The analysis explicitly shows that all models perform poorly on navigation. The extremely low F1-scores (~10%) are therefore likely dominated by poor exploration planning rather than a failure of visual hazard identification when the hazard is in view.
- The paper fails to sufficiently decouple the planning types of failures from the visual understanding failure. The low F1 score is a mixture of failures due to the robot not finding the objects (navigation failures) and acting upon finding the objects (perception/reasoning failures).
- Ambiguous Planning Strategy: The paper attributes the failure to a "significant weakness in the exploration effectiveness" of current embodied VLMs. However, the agent's action space consists only of basic navigation primitives.
- Insufficient Qualitative Discussion: The paper notes that as interaction turns grow, the precision rate significantly decreases, suggesting models start to generate false positives or hallucinate hazards. The discussion about this decay is insufficient, vaguely referencing "meaningless content". This crucial failure mechanism requires a deeper, qualitative understanding to diagnose why the performance degrades during long-horizon interaction.

**Questions:**

- How can the authors more effectively decouple the evaluation of the VLM's hazard identification ability (perception) from its embodied exploration planning ability? For instance, could the authors introduce a metric that evaluates hazard identification only on the subset of hazards that were correctly visible to the agent.
- In the analysis of multi-turn interaction, the Precision score significantly decreases over turns. What is the explicit, qualitative cause of this decrease?
- Why was VirtualHome chosen as the simulation environment over other platforms?
- The action space is limited to a small number of simple primitives. Would the results change significantly if the agent had access to a more diverse, high-level action space?
- The paper sets a maximum of 30 turns based on models outputting "meaningless content". This suggests a fundamental weakness in VLM long-term memory or planning. How useful even is using a VLM for these sorts of high-level navigation/embodied exploration tasks?

---

### Official Review · Reviewer_iDYo · 2025-10-31

**Soundness:** 2
**Presentation:** 3
**Contribution:** 2
**Rating:** 2
**Confidence:** 3

**Summary:**

This paper introduces HOMESAFEBENCH, a new benchmark for evaluating embodied Vision-Language Models (VLMs) in home safety inspection tasks. The authors identify two key limitations in prior work: an over-reliance on textual descriptions and the use of static, single viewpoints, which hinder accurate evaluation. HOMESAFEBENCH addresses this by providing 12,900 task instances in a 3D simulated environment. It requires agents to use first-person visual feedback and "free exploration" to identify five categories of common household hazards: fire, electric shock, falling object, trips, and child safety. The paper's main finding is that even top-tier VLMs (e.g., GPT-4O, Gemma3-12B) perform extremely poorly, with the best-performing model achieving an F1-score of only 10.23%. The authors conclude this demonstrates significant limitations in current VLM exploration and hazard identification capabilities.

**Strengths:**

- The paper is well-written, and the task is clearly described. Figures 1 and 2 effectively illustrate the agent's task flow and the data generation process. The core ideas are communicated effectively.

- The paper asks an important and significant question: are VLMs ready for real-world, embodied safety tasks? This problem formulation is valuable to the community.

**Weaknesses:**

**The "Free Exploration" Paradigm is Crippled by Design**: The paper's central contribution is "free exploration", but the action space provided to the agent makes meaningful exploration impossible and invalidates a key part of the benchmark.

- **Coarse Actions**: Actions are extremely coarse (e.g., 90-degree pivots, 3-step 'walks'). This is not exploration; it is a clunky grid-world navigation system that prevents fine-grained inspection. While this is a well-known limitation of traditional VLN tasks since 2018 (*1*) and the Virtual Home simulator (*2*), it’s important to note that the simulator already provides an angle of 30 degrees instead of 90 degrees.

- **Omission of "Look Down" Action**: The benchmark omits a "Look Down" action, which is a fatal flaw for a benchmark that explicitly tests for "Trip Hazards" (e.g., "a bar of soap left in a hallway"). These hazards are on the floor and are nearly impossible for the agent to perceive, making this category unsolvable through the provided "exploration" mechanism. Notably, the behavior of the "Look Down" action is not directly defined in the simulator, but it is trivial to construct. By binding the adjusted camera perspective to this action, this issue can be addressed (*reference here*).

---

**Conflated Metrics Offer No Diagnostic Value**: This is not a major weakness, but it is worth noting that the metrics used in the paper conflate at least four distinct failure modes, making it difficult to diagnose *why* models failed:

1. **Exploration Failure**: The agent never navigated to see the hazard.
2. **Perception Failure**: The agent saw the hazard but failed to identify it.
3. **Reasoning Failure**: The agent saw the hazard but failed to identify it as a *hazard*.
4. **Evaluation Failure**: The agent reasoned correctly but used a synonym not present in the evaluation map.

For instance, the case study in Figure 9, which claims a "misidentification," cannot distinguish between a perception failure and a reasoning failure.

---

**Comparison to Fixed Perspective (VLM) is Misleading**: Comparing to a fixed perspective (VLM) is not a valid baseline. The core claim in the paper is that "embodied exploration" is necessary. However, the benchmark never compares against a more direct "visual inspection" baseline. What would the performance of a VLM be if it were given a 360-degree panoramic image from the start position, with access to *zoom* and *crop* tools (such as o3 *reference here*) or other tool-oriented VLMs?

It would be interesting to compare the performance of this non-embodied agent against the paper's "exploration" agent. It’s possible that the "embodied" setup is merely an inefficient proxy for tool-use. My concern is that this type of agent can use tools to zoom in and crop pictures to see items more clearly or conduct a detailed analysis, effectively achieving a "walk-in" effect, similar to the intended exploration. The effects may be consistent in most scenarios.

**Questions:**

See weakness.

---

### Official Review · Reviewer_4YaQ · 2025-11-01

**Soundness:** 2
**Presentation:** 2
**Contribution:** 2
**Rating:** 2
**Confidence:** 4

**Summary:**

The manuscript proposes a new benchmark for evaluating embodied AI agents’ abilities to identify and report safety hazards in home environments, according to five pre-defined safety hazard categories, via first-person perspective images under free exploration.

**Strengths:**

- The manuscript is well-written and well-organized
- The manuscript considers compelling tasks in Embodied AI
- The manuscript provides a reasonable set of initial experiments that provide headroom for subsequent research

**Weaknesses:**

- Especially because the benchmark features heavily photo-synthetic visual environments, the manuscript should provide some error analysis on the failure cases. This analysis would help to distinguish between cases where the models fail to reason effectively about the ramifications of the potentially hazardous situation, versus cases where the models are simply unable to identify the semantically relevant object, etc.
- Lacking comprehensive discussion on the prompt scheme design. After reviewing the prompts, the manuscript seems to be forcing the models to perform classification tasks, as opposed to reasoning deeply. Missing a principled discussion and experiments on various prompting strategies, e.g., text-only and vision-based Chain-of-Thought, decomposition-based prompting, self-consistency / self-refinement, etc.

**Questions:**

Nothing additional; please see comments above.

---

### Note · Authors · 2025-11-13

**Comment:**

We sincerely thank you for your **diligent work and constructive feedback**. We appreciate your positive comments regarding our motivation and dataset construction. We also acknowledge the valuable points raised concerning our agent interaction design flaws and the need for a more detailed qualitative discussion.

We have decided to withdraw the manuscript at this time. We will thoroughly incorporate your valuable feedback for an improved version in the future.

**Thank you once again for your time and expertise.**

**Withdrawal Confirmation:**

I have read and agree with the venue's withdrawal policy on behalf of myself and my co-authors.